# Green Tea Catechins Trigger Immediate-Early Genes in the Hippocampus and Prevent Cognitive Decline and Lifespan Shortening

**DOI:** 10.3390/molecules25071484

**Published:** 2020-03-25

**Authors:** Keiko Unno, Monira Pervin, Kyoko Taguchi, Tomokazu Konishi, Yoriyuki Nakamura

**Affiliations:** 1Tea Science Center, University of Shizuoka, 52-1 Yada, Suruga-ku, Shizuoka 422-8526, Japan; gp1747@u-shizuoka-ken.ac.jp (M.P.); gp1719@u-shizuoka-ken.ac.jp (K.T.); yori.naka222@u-shizuoka-ken.ac.jp (Y.N.); 2Faculty of Bioresources Sciences, Akita Prefectural University, Shimoshinjo Nakano, Akita 010-0195, Japan; konishi@akita-pu.ac.jp

**Keywords:** green tea catechin, cognitive function, immediate-early gene, lifespan, SAMP10

## Abstract

Senescence-accelerated mouse prone 10 (SAMP10) mice, after ingesting green tea catechins (GT-catechin, 60 mg/kg), were found to have suppressed aging-related decline in brain function. The dose dependence of brain function on GT-catechin indicated that intake of 1 mg/kg or more suppressed cognitive decline and a shortened lifespan. Mice that ingested 1 mg/kg GT-catechin had the longest median survival, but the dose was less effective at suppressing cognitive decline. The optimal dose for improving memory acquisition was 60 mg/kg, and memory retention was higher in mice that ingested 30 mg/kg or more. To elucidate the mechanism by which cognitive decline is suppressed by GT-catechin, changes in gene expression in the hippocampus of SAMP10 mice one month after ingesting GT-catechin were analyzed. The results show that the expression of immediate-early genes such as nuclear receptor subfamily 4 (*Nr4a*), FBJ osteosarcoma oncogene (*Fos*), early growth response 1 (*Egr1*), neuronal PAS domain protein 4 (*Npas4*), and cysteine-rich protein 61 (*Cyr61*) was significantly increased. These results suggest that GT-catechin suppresses age-related cognitive decline via increased expression of immediate-early genes that are involved in long-term changes in plasticity of synapses and neuronal circuits.

## 1. Introduction

Senescence-accelerated mouse prone 10 (SAMP10) mice that ingested green tea catechins (GT-catechin, 60 mg/kg) were found to have suppressed aging-related decline in brain function [1,2]. SAMP10 mice used in this study had characteristic accelerated senescence and age-related cognitive decline [3,4]. Since SAMP10 mice generate more superoxide anions, a reactive oxygen species (ROS), than SAMR1 mice, which have the same genetic background but a normal lifespan, oxidative damage is considered to be a factor that promotes brain aging [5]. In fact, mice that consumed GT-catechin at this dose had increased activity of glutathione peroxidase and reduced brain peroxidation [6,7]. However, ROS generation was not suppressed even in mice that consumed GT-catechin (unpublished data). GT-catechin may also suppress brain aging via another pathway other than the suppression of oxidative damage.

Since the daily concentration of GT-catechin ingested by Japanese people is about 0.3–0.6 mg/mL, for animal experiments a dose of about one-third that concentration was tried, considering the effects of species differences. When SAMP10 mice (average body weight 33 g) drank 10 mL/day of water containing GT-catechin (0.2 mg/mL), catechin intake was 60 mg/kg. GT-catechin contains epigallocatechin gallate (EGCG) as the main catechin, followed by epigallocatechin (EGC). We found that EGCG is important for suppression of cognitive decline [7]. In addition, learning time was significantly shorter in mice that ingested EGCG for 5 months and more than in age-matched control mice, while the difference in starting age of EGCG ingestion had little effect on learning ability [8]. A shorter ingestion period of two or three months tended to suppress the decrease in learning ability [8]. It seems that catechin intake needs to be continued for a certain length of time.

Furthermore, we speculated that EGCG could be absorbed into the brain parenchyma via the blood–brain barrier, thereby promoting neuronal differentiation [7]. However, since the blood–brain barrier permeability of EGCG is about 4% and the concentration of EGCG in the brain is considerably lower than in the periphery, the effects of catechins may be different in the brain and in the periphery. In this study, we investigated the effects of GT-catechin on cognitive function and longevity at concentrations of 1–60 mg/kg.

The effects of flavonoids containing EGCG on lifespan in nematodes, flies, and mice have been reviewed [9]. Studies with mutant nematodes and flies gave us new information about genes regarding longevity regulated by catechins. Xiong et al. reported that EGCG promotes nematode longevity by inducing ROS production and triggering mitochondrial biosynthesis [10]. Mice have longer lifespans than nematodes and flies, and therefore take a longer time to study, but can provide important evidence to examine the effects of catechins on the human lifespan.

Regarding the effects of catechins on cognitive function, improvements have been reported in Alzheimer’s disease model mice [11,12,13,14], Down syndrome model mice [15], high-fat and high-fructose diet ingesting mice [16], a rat model with reduced cerebral blood flow [17], mice with stress-induced cerebral dysfunction [18], a streptozotocin-induced model of dementia [19], and post-traumatic brain injury [20]. On the other hand, there are some reports that EGCG intake did not affect cognitive function, for example, in mice that had induced inflammation [21], in a Down syndrome mouse model [22], and in aged mice that were given beta-alanine [23]. Where does this difference come from? The antioxidative activity of catechins has been explored for their protective effects in a variety of systems [24]. In addition, neurogenesis in the adult hippocampus is critically involved in adult brain function, and EGCG has been reported to promote adult hippocampal neurogenesis [25]. Furthermore, it has been reported that EGCG plays an important role in the development of the nervous system, in forming connections between neurons [26]. Therefore, we comprehensively investigated gene expression changes in the hippocampus of mice ingesting catechins.

## 2. Results

### 2.1. Effect of GT-Catechin Ingestion on Lifespan

SAMP10 mice are susceptible to aging and have a shorter lifespan than SAMR1 and other mouse strains. Although the median survival time (MST) was 10.8 months under conventional conditions, it was significantly prolonged in mice that ingested more than 1 mg/kg of GT-catechin (Figure 1). The longest MST was observed in mice that ingested GT-catechin at 1 mg/kg (Table 1). This was 1.59 times longer than the MST of control mice.

MST was 15.3 months in mice that ingested GT-catechin at a concentration range of 5–30 mg/kg. This was 1.42 times longer than the MST of control mice. In mice that ingested GT-catechin at 60 mg/kg, MST was 1.26 times longer than that of control mice.

### 2.2. Memory Acquisition, Memory Retention, and Working Memory

To evaluate memory acquisition, the time for learning not to enter a dark room was measured when mice were 11 months old using a step-through passive avoidance task. A longer learning time implies lower learning ability. The dose-dependency of GT-catechin was examined. Learning time was significantly shorter in mice that ingested 1 mg/kg or more of GT-catechin than in control mice that ingested no GT-catechin (Figure 2). The time for learning was shortest in mice that ingested 15 mg/kg of GT-catechin.

Memory retention ability was measured at 12 months of age. To evaluate memory retention, mice were examined by the step-through passive avoidance task 1 month later. Long-term memory was better in mice that ingested a higher concentration of GT-catechins. The ratio of mice that consumed 30 mg/kg of GT-catechin tended to be higher compared to control mice (*p* = 0.060), and the ratio of mice that consumed 60 mg/kg of GT-catechin was significantly higher compared to control mice (*p* = 0.024) (Table 2).

The working memory of 11-month-old SAMP10 mice was examined using a Y-maze. This test is based on the natural instinct of a mouse to investigate a new area rather than one that it has just searched. This spontaneous alternative behavior is an index of working memory. When the searching behavior of the mice was observed, no effect was found in mice that ingested less than 5 mg/kg of GT-catechin. However, the spontaneous alternative ratio increased in a dose-dependent manner in mice that ingested more than 15 mg/kg of GT-catechin. A significant effect was observed in mice that ingested 30 mg/kg or more of GT-catechin (Figure 3).

### 2.3. Transcriptome

The hippocampi of two-month-old mice that ingested GT-catechin for one month were used for analysis. DNA microarray data of GT-catechin, which were obtained using high-density oligonucleotide microarrays, showed 605 positive genes based on two-way ANOVA (*p* < 0.001). The top 20 genes that were significantly upregulated following the ingestion of GT-catechin are listed in Table 3. Growth hormone (Gh) is synthesized in the central nervous system and is critical for neural development [27]. Among these 20 genes, early growth response 2 (Egr2), activity regulated cytoskeletal-associated protein (Arc), nuclear receptor subfamily 4, groupA, member 1 (Nr4a1), FBJ osteosarcoma oncogene (Fos), neuronal PAS domain protein 4 (Npas4), early growth response 1 (Egr1), and cysteine rich protein 61 (Cyr61) are immediate-early genes (IEGs) [28,29,30]. Dual specifificity phosphatase 1 (Dusp1) is reported to be an IEG in sensory neurons [31]. GTP-binding protein (Gem) promotes morphological differenciation in neurons [32]. Heat shock protein 1A (Hspa 1a) and heat shock protein 1 (Hspb1) are involved in stress and cell differentiation [33,34]. Cysteine-rich EGF-like domains 2 (Creld2) and stromal cell-derived factor 2-like 1 (Sdf2l1) play a role in endoplasmic reticulum (ER) homeostasis [35,36]. Period homolog 1 (Per1), a circadian clock gene, is critical for long-term memory formation [37].

### 2.4. Effect of GT-Catechin Intake on the IEG Levels

The increase in IEGs (*Egr2*, *Arc*, *Nr4a1*, *Fos*, *Npas4*, *Egr1i* and *Cyr61*) was confirmed by qRT-PCR. The degree of gene expression in the hippocampus and prefrontal cortex of mice that ingested GT-catechin for one, five, or 11 months was compared with control mice of the same age (two, six, and 12 months of age, respectively). The expression levels of IEGs (*Nr4a1*, *Fos*, *Npas4*, *Egr1*, and *Cyr61*) increased significantly in the hippocampi of two-month-old mice that ingested GT-catechin, but the difference was lower in six-month-old mice and was not observed in 12-month-old mice (Figure 4). *Egr2* was not significantly increased (*p* = 0.08) but tended to be higher in two-month-old mice that ingested GT-catechin compared to age-matched controls. *Arc* was not significantly higher than the age-matched controls. In the prefrontal cortex, effects of GT-catechin ingestion and aging on the expression of IEGs were not observed (data are not shown). Although the increase of *Gh* expression following catechin intake was interesting, there were large individual differences in the expression levels of *Gh* (data not shown).

## 3. Discussion

We examined how much GT-catechin intake is needed to prevent age-related cognitive decline. After examining the learning (memory acquisition) ability by a step-through passive avoidance task, the results showed that the intake of GT-catechin at 1 mg/kg or more showed a significant improvement, and that mice that consumed 15 mg/kg of GT-catechin had the best results (Figure 2). Long-term memory (memory retention) ability, as assessed by the step-through passive avoidance task performed one month later, increased with increasing concentrations of GT-catechin, with a significant improvement observed when 60 mg/kg of GT-catechin was consumed (Table 2). Working memory, as evaluated by the alternative ratio using a Y-maze, increased significantly when 30 mg/kg or more of GT-catechin was consumed (Figure 3). Taken together, these results suggest that GT-catechin intake of at least 1 mg/kg or more suppresses aging-related cognitive decline in SAMP10 mice.

MST increased significantly (1.6-fold) when 1 mg/kg of GT-catechin was consumed relative to the control group, and increased 1.3-fold when 60 mg/kg was consumed. High doses of GT-catechin are not required to suppress the shortening of longevity in SAMP10 mice. There was no change in maximum lifespan. Antioxidants can control autoxidation, reduce oxidative stress, and subsequently increase healthy longevity [38]. In fact, oxidative stress (the level of 8-oxodeoxyguanosine) was significantly lower in the liver and kidney of mice that consumed GT-catechin than age-matched control mice (data not shown). However, the effects on longevity may not be fully explained by antioxidant activity alone, as mice that ingested 1 mg/kg GT-catechin rather than 60 mg/kg had a longer lifespan. Since catechin has both antioxidant and pro-oxidant properties [39], GT-catechin may promote mouse longevity as in nematodes, by inducing ROS production and triggering mitochondrial biosynthesis [10]. In addition, it has been suggested that 4%–5% of GT-catechin in the blood reaches the brain parenchyma via the blood–brain barrier [40]. Therefore, differences in brain and peripheral GT-catechin levels can have different effects on cognitive function and longevity in SAMP10 mice. Further investigation is needed to elucidate the mechanism of GT-catechin in terms of its effects on longevity.

In any case, this study revealed that intake of GT-catechin at a dose of 1 mg/kg or more per day tends to improve both age-related cognitive decline and lifespan shortening. Whether this dose is directly applicable to humans is not yet known. However, since GT-catechin in green tea eluate is 30–60 mg/100 mL, drinking at least 1–2 cups of green tea every day might benefit the suppression of aging-related cognitive decline and lifespan shortening. Indeed, some epidemiological studies have shown that consuming green tea daily reduces the risk of dementia [41], and frequent consumption of green tea (≤5 cups/day) reduces the risk of developing dementia [42]. A study of the relationship between green tea intake and the risk of mortality in Japanese men and women showed that the risk of mortality decreases as green tea consumption increases [43].

Next, the mechanism to suppress age-related cognitive decline in mice that ingested GT-catechin was examined. Studies using cultured neurons suggest that EGCG has a stronger effect on neurite outgrowth than EGC [7]. Neurons transmit information by expanding neurites and forming synapses, therefore, the effect of EGCG on neurite outgrowth is noteworthy. To clarify the target of GT-catechin in the brain, we examined changes in gene expression that occurred in the hippocampus of mice that ingested GT-catechin.

DNA microarray and qRT-PCR results indicate that the expression of some IEGs (*Nr4a1*, *Fos*, *Egr1*, *Npas4*, and *Cyr61*) was significantly increased in two-month-old mice that ingested GT-catechin for one month. Increased expression of three of these IEGs (Fos, Egr-1, and Npas4) plays a key role in long-term synaptic plasticity [44]. In addition, Npas4 is thought to regulate excitatory and inhibitory balance within circuits. Nr4a1 is a key component that regulates the density and distribution of spines and synapses [45], and has been reported to be involved in the suppression of age-related decline in brain function [46]. Furthermore, Cyr61 is needed for the dendritic arborization of hippocampal neurons [30]. Increased expression of these IEGs is considered to increase synaptic plasticity, leading to the maintenance and improvement of learning and memory ability. IEGs are endogenous genes whose expression is first induced in response to extracellular stimuli, and their expression is widely used as a marker of neural activity. Increased expression of IEGs was more pronounced in the hippocampus than in the prefrontal cortex, suggesting that it is important for hippocampal function. The transcription of many IEGs in neurons is initiated by calcium ion influx associated with synaptic activity and action potential [47]. It has also been reported that EGCG modulates calcium signals in hippocampal neurons [48,49]. These data suggest that EGCG incorporated into the hippocampus stimulates IEG expression by causing an increase in intracellular calcium ions in nerve cells.

In conclusion, GT-catechin, mainly EGCG, suppressed age-related cognitive decline and lifespan shortening. The expression of some IEGs was increased in the hippocampi of mice that ingested GT-catechin. The common factor that causes increased expression of these IEGs may be an increase in calcium ions in neurons triggered by EGCG.

## 4. Materials and Methods

### 4.1. Animals

Four-week-old male SAMP10/TaSlc (SAMP10) mice were purchased from Japan SLC Co. Ltd. (Shizuoka, Japan) and kept in conventional conditions in a temperature- and humidity-controlled room with a 12 h–12 h light–dark cycle (light period, 08:00–20:00; temperature, 23 ± 1 °C; relative humidity, 55 ± 5%). Mice were fed with a normal diet (CE-2; Clea Co. Ltd., Tokyo, Japan) and water ad libitum. Mice were housed in groups of 6 per cage. All experimental protocols were approved by the University of Shizuoka Laboratory Animal Care Advisory Committee (approval no. 136068) and were in accordance with the guidelines of the US National Institutes of Health for the care and use of laboratory animals.

### 4.2. Experimental Design

For the experiment, 180 mice were prepared and divided into 6 groups containing a control. Six mice were housed per cage. These mice were used for measurement of cognitive function and longevity. Three groups of 36 mice each consumed GT-catechin (Sunphenone BG; Taiyo Kagaku Co. Ltd., Yokkaichi, Japan) in water at concentrations of 0, 50, and 200 µg/mL from 1 month of age. These mice (body weight 30–35 g) drank about 10 mL of water daily. This intake of GT-catechin corresponds to 0, 15, and 60 mg/kg, respectively. Another 3 groups of 24 mice each consumed GT-catechin in water at concentrations of 3.3, 17, and 100 µg/mL from 1 month of age. These concentrations correspond to 1, 5, and 30 mg/kg, respectively. Mice consumed GT-catechin solution, which was freshly prepared twice a week. The amount of GT-catechin water consumed by mice was measured and the GT-catechin intake was calculated.

Another 48 mice were used for DNA microarray analysis and quantitative real-time reverse transcription PCR (qRT-PCR). The mice consumed GT-catechin in water at a concentration of 200 µg/mL (60 mg/kg). Sunphenone BG contains several catechins: 40.7 w/w% EGCG, 17.4 w/w% EGC, 12.3 w/w% ECG, 7.6 w/w% EC, 3.1 w/w% gallocatechin, 1.9 w/w% catechin, and 1.7 w/w% gallocatechin gallate. The remaining portion consists of some other catechins from green tea. Sunphenone BG does not contain caffeine.

### 4.3. Memory Acquisition and Retention Tests

A step-through passive avoidance task was carried out using 11-month-old mice as described previously [1,2]. In brief, when a mouse entered a dark chamber from a light chamber, the door was closed and an electric foot-shock was delivered at 50 µA for 1 s (SGS-003, Muromachi Kikai Co., Ltd., Tokyo, Japan). Acquisition of the avoidance response was judged as successful if the mouse remained in the light chamber for 300 s. The trial was repeated until the mouse satisfied the acquisition criterion within 5 trials. The time that a mouse could not stay in the light chamber, i.e., the time spent in the dark chamber in a 300 s trial, was recorded. The results from successive trials were summed for each mouse to give a measure of the time required to learn not to enter the light chamber (i.e., “learning time”). One month later, the same test was performed on the mice. If the mouse was able to stay in the light chamber for 300 s, retention of the avoidance response was determined to be successful. The ratio was obtained as memory-retained mice/tested mice.

### 4.4. Working Memory

Searching behavior was observed in a Y-maze (MYM-01M; Muromachi Kikai Co., Ltd.). The positions of arms that a mouse entered and the number of times it entered them was observed for 8 min, as described previously [1]. “Alternation behavior,” i.e., entering the 3 different arms successively, is considered to reflect working memory capacity. The number of occasions on which spontaneous alternation behavior was observed was counted and the “ratio of alternation” was calculated as follows: (number of arm entries showing spontaneous alternation)/(total number of arm entries - 2).

### 4.5. Measurement of DNA Microarray and Principal Component Analysis

The mice were housed in groups of 6 for 1 month and ingested water containing green tea catechin or nothing (control). Every three mice 2 months of age were anesthetized with isoflurane and blood was removed from the jugular vein. The hippocampus was removed and frozen immediately. Total RNA was extracted from the hippocampus using an RNeasy Mini Kit (74104, Qiagen, Valencia, CA, USA). Total RNA was processed to synthesize biotinylated cRNA using One-Cycle Target Labeling and Control Reagents (Affymetrix, Santa Clara, CA, USA) and then hybridized to a Total RNA Mouse Gene 1.0 ST Array (Affymetrix), with 3 biological repeats per group. Raw data were parametrically normalized [50] by using the SuperNORM data service (Skylight Biotech Inc., Akita, Japan). The significance of GT-catechin ingestion was statistically tested by two-way ANOVA [51] at *p* < 0.001.

To compare the effects of GT-catechin ingestion, we performed principal component analysis (PCA) [52] on ANOVA-positive genes [53]. To reduce the effects of individual variability among samples, the axes of PCA were estimated on a matrix of each group’s sample means and applied to all data, which were centered using the sample means of control mice.

### 4.6. Quantitative Real-time Reverse Transcription PCR (qRT-PCR)

Mice 2, 6, and 12 months of age that ingested water containing GT-catechin (60 mg/kg) or not were used for this analysis. Mice were anesthetized with isoflurane and blood was removed from the jugular vein. The brain was carefully dissected and the hippocampus and prefrontal cortex were immediately frozen. The brain sample was homogenized, and total RNA was isolated using a purification kit (NucleoSpin^®®®^ RNA, 740955, TaKaRa Bio Inc., Shiga, Japan) in accordance with the manufacturer’s protocol. The obtained RNA was converted to cDNA using the PrimeScript^®®®^ RT Master Mix kit (RR036A, Takara Bio Inc.). qRT-PCR analysis was performed using the PowerUp™ SYBR™ Green Master Mix (A25742, Applied Biosystems Japan Ltd., Tokyo, Japan) and automated sequence detection systems (StepOne, Applied Biosystems Japan Ltd.). Relative gene expression was measured by previously validated primers for *Egr2* [54], *Arc* [55], *Nr4a1* [56], *Fos* [57], *Egr1* [58], *Npas4* [59], and *Cyr61* [60] genes (Table 4). cDNA derived from transcripts encoding β-actin was used as the internal control.

### 4.7. Statistical Analysis

Statistical analysis for cognition activity was performed using one-way ANOVA. Confidence intervals and significance of differences in means were estimated by using Tukey’s honest significant difference method. Fisher’s exact probability test was used for qRT-PCR. After calculating the survival rate by the Kaplan–Meier method, the difference in survival rate was tested by the log rank test.

## 5. Conclusions

This study reveals that intake of GT-catechin at a dose of 1 mg/kg or more per day tends to improve both age-related cognitive decline and lifespan shortening. GT-catechin, mainly EGCG, suppressed age-related cognitive decline via the increased expression of some IEGs in the hippocampus.

## Figures and Tables

**Figure 1 molecules-25-01484-f001:**
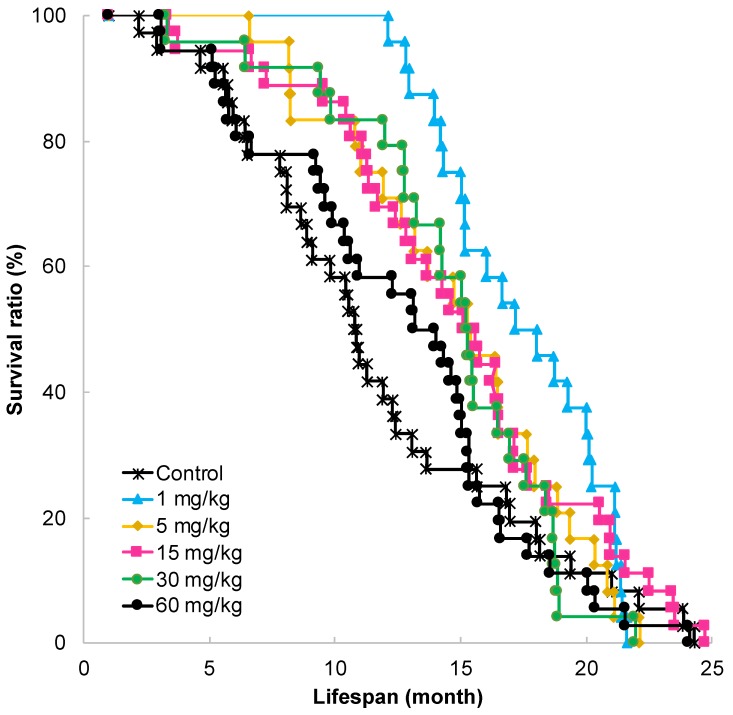
Effect of catechin ingestion on longevity of SAMP10 mice that consumed GT-catechin in water from 1 month of age. GT-catechin solution was freshly prepared twice a week. Three groups of 36 mice each consumed 0 (control), 15, and 60 mg/kg of GT-catechin. Another 3 groups of 24 mice each consumed 1, 5, and 30 mg/kg of GT-catechin.

**Figure 2 molecules-25-01484-f002:**
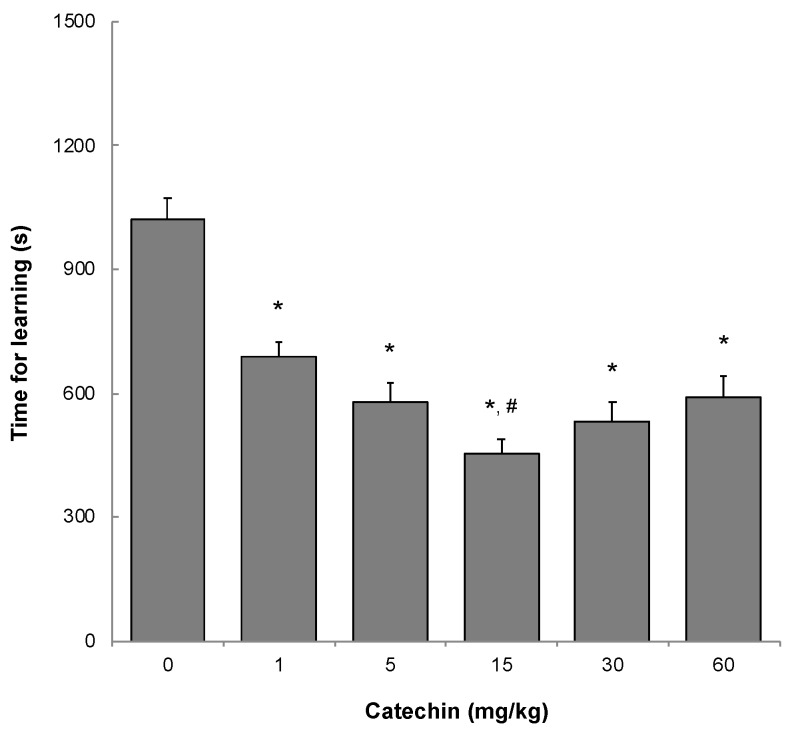
Effect of GT-catechin ingestion on learning ability of SAMP10 mice. A step-through passive avoidance task was carried out using 11-month-old mice. When a mouse entered a dark chamber from a light chamber, the door was closed and an electric foot-shock was delivered at 50 µA for 1 s. Acquisition of the avoidance response was judged as successful if the mouse remained in the light chamber for 300 s. The trial was repeated until the mouse satisfied the acquisition criterion within five trials. This result from successive trials was summed for each mouse to give a measure of the time required for learning not to enter the light chamber (i.e., learning time) (n = 16–24; * *p* < 0.05 to control, # *p* < 0.05 to mice that ingested 1 mg/kg).

**Figure 3 molecules-25-01484-f003:**
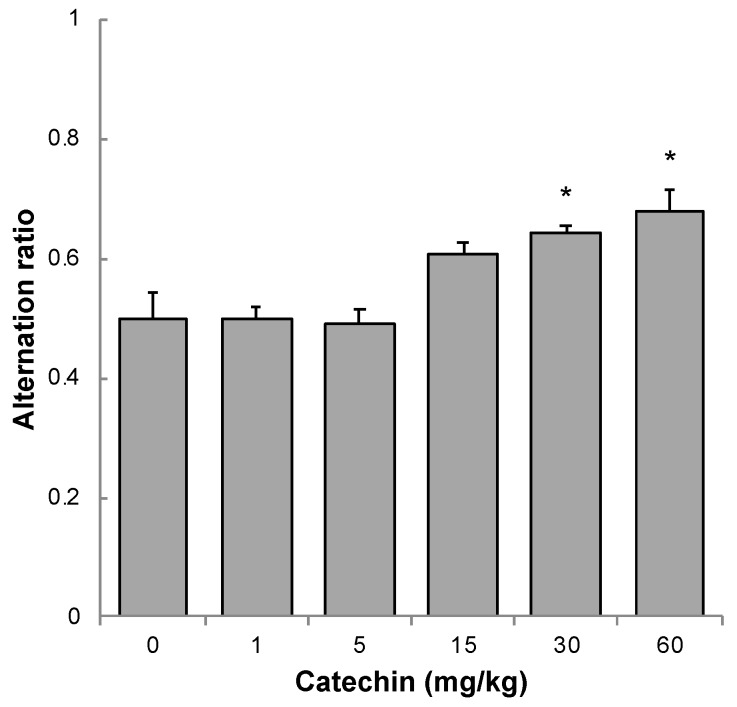
Effect of GT-catechin ingestion on working memory of 11-month-old SAMP10 mice. Searching behavior was observed in a Y-maze. The number of occasions in which spontaneous alternation behavior was observed was counted and the ratio of alternation was calculated as follows: (number of arm entries showing spontaneous alternation)/(total number of arm entries − 2) (n = 18–21; * *p* < 0.05)

**Figure 4 molecules-25-01484-f004:**
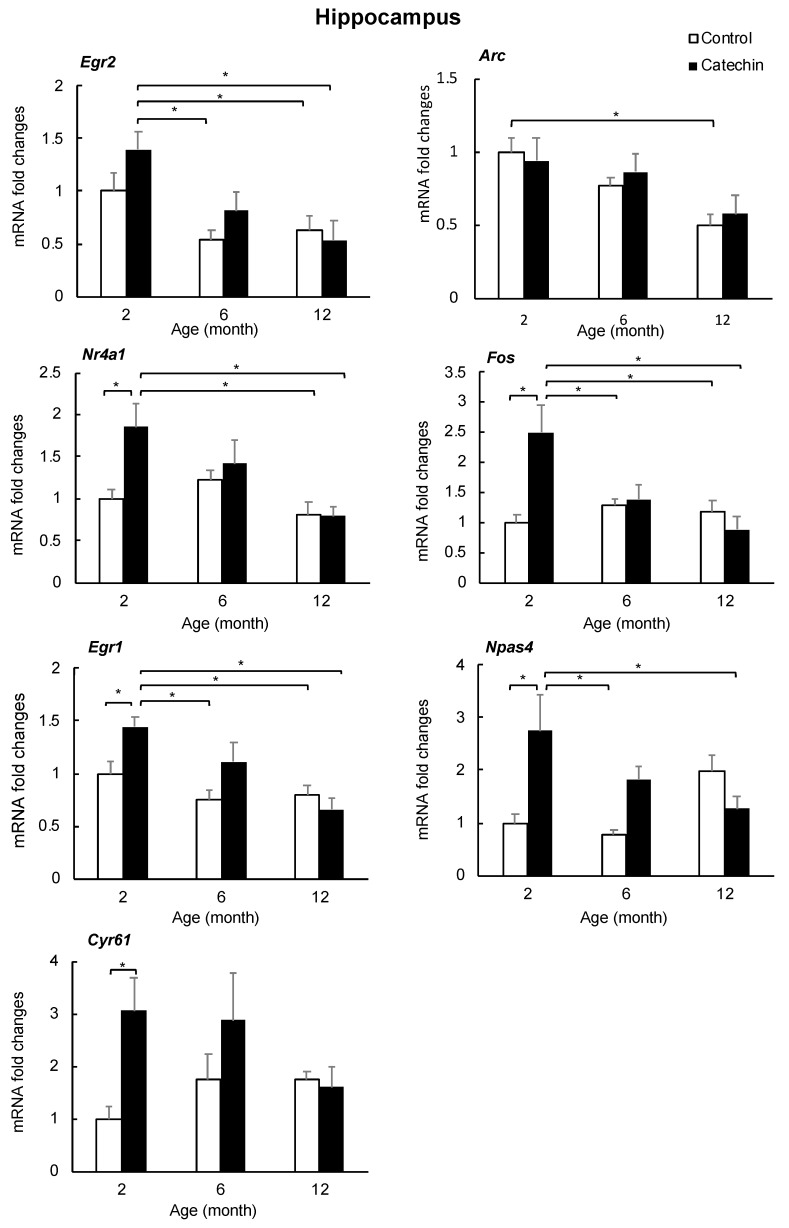
Expression of immediate-early genes (IEGs) in hippocampi of mice that ingested GT-catechin (60 mg/kg) and controls (n = 6, * *p* < 0.05).

**Table 1 molecules-25-01484-t001:** Effect of green tea catechin (GT-catechin) ingestion on median survival time (MST) of senescence-accelerated mouse prone 10 (SAMP10) mice.

GT-Catechin	MST (Months)		*p*-Value
(mg/kg)	Month	Ratio	
0	10.8	1.00	-
1	17.2	1.59	0.027 *
5	15.3	1.42	0.272
15	15.3	1.42	0.082
30	15.3	1.42	0.364
60	13.6	1.26	0.880

*p*-value is based on log-rank test. * *p* < 0.05.

**Table 2 molecules-25-01484-t002:** Effect of catechin ingestion on long-term memory of 12-month-old SAMP10 mice.

	Ratio	*p*-Value
0	0.567	—
1	0.583	0.533
5	0.720	0.186
15	0.737	0.185
30	0.800	0.060
60	0.833	0.024 *

One month after the step-through passive avoidance task, the same test was performed on the mice. If a mouse was able to stay in the light chamber for 300 s, then memory was determined to have been retained. The ratio represents memory-retained mice/tested mice (n = 19–36; *, *p* < 0.05).

**Table 3 molecules-25-01484-t003:** Upregulated genes in hippocampi of mice that ingested GT-catechin (60 mg/kg): top 20.

Symbol.	Full Name	ΔZ	*p*
Gh	growth hormone	0.5621	4.55 × 10^−7^
Egr2	early growth response 2	0.3793	1.32 × 10^−24^
Arc	activity regulated cytoskeletal-associated protein	0.2996	2.49 × 10^−35^
Nr4a1	nuclear receptor subfamily 4, group A, member 1	0.2858	1.79 × 10^−37^
Fos	FBJ osteosarcoma oncogene	0.2497	4.31 × 10^−20^
Egr1	early growth response 1	0.2216	1.56 × 10^−28^
Dusp1	dual specificity phosphatase 1	0.2123	1.07 × 10^−23^
Gem	GTP binding protein (gene overexpressed in skeletal muscle)	0.1948	1.21 × 10^−12^
Hspa1a	heat shock protein 1A	0.1916	1.28 × 10^−11^
Rtl1	retrotransposon-like 1	0.1847	8.34 × 10^−8^
Hspa1a	heat shock protein 1A	0.1827	8.76 × 10^−22^
Hspb1	heat shock protein 1	0.1795	3.10 × 10^−9^
Npas4	neuronal PAS domain protein 4	0.1735	7.10 × 10^−12^
Cyr61	cysteine rich protein 61	0.1687	1.93 × 10^−10^
Creld2	cysteine-rich with EGF-like domains 2	0.1674	1.34 × 10^−14^
Per1	period homolog 1 (Drosophila)	0.1671	5.43 × 10^−15^
Unc13c	unc-13 homolog C (C. elegans)	0.1559	4.13 × 10^−5^
Hey2	hairy/enhancer-of-split related with YRPW motif 2	0.1551	1.21 × 10^−9^
Agxt2l	alanine-glyoxylate aminotransferase 2-like 1	0.1526	0.000375
Sdf2l1	stromal cell-derived factor 2-like 1	0.1503	1.09 × 10^−8^
	ΔZ = expression level (catechin–control)		

**Table 4 molecules-25-01484-t004:** Primer sequences for qRT-PCR.

Gene	Forward Sequence (5′-3′)	Reverse Sequence (5′-3′)	Ref.
*Egr2*	CTACCCGGTGGAAGACCTC	AATGTTGATCATGCCATCTCC	[54]
*Arc*	ACGATCTGGCTTCCTCATTCTGCT	AGGTTCCCTCAGCATCTCTGCTTT	[55]
*Nr4a1*	CTGCCTTCCTGGAACTCTTCA	CGGGTTTAGATCGGTATGCC	[56]
*Fos*	AAGTAGTGCAGCCCGGAGTA	CCAGTCAAGAGCATCAGCAA	[57]
*Egr1*	CCTTCCAGTGTCGAATCTGCAT	ACAAATGTCACAGGCAAAAGGC	[58]
*Npas4*	AGCATTCCAGGCTCATCTGAA	GGCGAAGTAAGTCTTGGTAGGATT	[59]
*Cyr61*	CCCCCGGCTGGTGAAAGTC	ATGGGCGTGCAGAGGGTTGAAAAG	[60]

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
