# Peer review of "Green Tea Catechins Trigger Immediate-Early Genes in the Hippocampus and Prevent Cognitive Decline and Lifespan Shortening"

_molecules, 2020, doi:10.3390/molecules25071484_

Round 1
Reviewer 1 Report
This paper by Unno and collaborators reports results on the effect of green tea catechins on cognitive decline and lifespan shortening. They used senescence-accelerated mouse prone 10 line as a model organism and provided data on differential gene expression pattern for that mice feed with green tea catechins compared to controls.
Overall, the manuscript is well written, but I strongly believe that some point needs to be carefully reviewed in order to consider it for publication.
Major point
The authors reported two types of analysis, in one case, they analyzed effects on lifespan and cognitive performance, and in other case, they provided data at molecular level by differential gene expression analysis by means of microarray and qPCR approaches. Since qPCR results sound very important in the paper (the authors highlighted these results in the title) it is mandatory to carefully check the primers used in that approach. The primers sequences are reported in the table 4 of materials and methods and after extensive research in nucleotide database (NCBI), I was unable to find correspondence between primers sequences and the nucleotide sequence of the corresponding analyzed transcript for Arc gene. In particular, the authors used primers reported in the reference 37, but in that paper, the primers were designed for rat and not for mouse. In fact, searches by primer blast were unable to find correspondence of that primers for mouse transcript. Even if I manually analyses the transcript sequence of mouse Arc, I was able to find the corresponding sequences. I strongly believe that the authors have to check the primers sequence and have to report the accession number of the transcript sequence in order to give the opportunity to find the correspondence.
Minor point
- In my opinion the introduction appeared too short and should be extended including other papers on the effect of green tea catechins on the brain development and function.
- In the introduction, they reported the sentence “We found that EGCG is important”. The authors should specify “important” for what.
- In the introduction, the following two sentences lack of reference.
“A shorter ingestion period of 2 and 3 months tended to suppress the decrease in learning ability.”
“The ingestion of EGCG may cause some changes in the brain”.
Without reference, the reader may interpret that those sentences are related to experimental results described in the current paper and not data from the state of the art reported in literature.
- In the introduction, the authors should specify what they mean for “changes in the brain” in the sentence “The ingestion of EGCG may cause some changes in the brain”.
- In the results section, in my opinion, the authors should first report the effect of GT-catenins on lifespan and then data on memory acquisition, memory retention and working memory. This is especially important considering that they stated that SAMP10 mice have a median survival time of 10.8 months and they analyzed cognitive performance in 11-mont-old mice.
- In the results section (paragraph 2.3), the authors reported transcriptomic analysis to find differentially expressed gene in the hippocampus of SAMP10 mice feed with GT-catenins compared to controls. I was unable to find the indication of mice age as clearly reported for the qPCR analysis. This data is important to compare results from microarray and qPCR approach.
- In the results section (paragraph 2.4), the authors reported that “The increase in IEGs was confirmed by qRT-PCR”. At the same time, they did not confirm the gene expression variation for EGR2 (the transcript with the highest expression variation).
- In the figure 4, the panel related to Cyr61 contains the indication of control that needs to be removed.
- The figure 5 may be removed considering that no variation has been demonstrated. That data might be reported as “data not shown”.
Author Response
Response to Reviewer 1
This paper by Unno and collaborators reports results on the effect of green tea catechins on cognitive decline and lifespan shortening. They used senescence-accelerated mouse prone 10 line as a model organism and provided data on differential gene expression pattern for that mice feed with green tea catechins compared to controls.
Overall, the manuscript is well written, but I strongly believe that some point needs to be carefully reviewed in order to consider it for publication.
Major point
The authors reported two types of analysis, in one case, they analyzed effects on lifespan and cognitive performance, and in other case, they provided data at molecular level by differential gene expression analysis by means of microarray and qPCR approaches. Since qPCR results sound very important in the paper (the authors highlighted these results in the title) it is mandatory to carefully check the primers used in that approach. The primers sequences are reported in the table 4 of materials and methods and after extensive research in nucleotide database (NCBI), I was unable to find correspondence between primers sequences and the nucleotide sequence of the corresponding analyzed transcript for Arc gene. In particular, the authors used primers reported in the reference 37, but in that paper, the primers were designed for rat and not for mouse. In fact, searches by primer blast were unable to find correspondence of that primers for mouse transcript. Even if I manually analyses the transcript sequence of mouse Arc, I was able to find the corresponding sequences. I strongly believe that the authors have to check the primers sequence and have to report the accession number of the transcript sequence in order to give the opportunity to find the correspondence.
This was our serious mistake. We greatly appreciate for your finding the mistake. We carefully performed qRT-PCR using the new probe. The result showed that there was no significant difference in the expression level of Arc between 2-month-old mice ingested GT-catechin and the age-matched control. DNA microarray data indicated that the expression was significantly increased in the GT-catechin group, but some mouse samples used for qRT-PCR showed slightly lower values than the control. Unfortunately, we were unable to obtain similar data for Arc between the DNA microarray and qRT-PCR. Figure 4 and Table 4 were changed.
Minor point
- In my opinion the introduction appeared too short and should be extended including other papers on the effect of green tea catechins on the brain development and function.
We added some other papers on the effect of green tea catechins on cognitive function and longevity in the introduction.
- In the introduction, they reported the sentence “We found that EGCG is important”. The authors should specify “important” for what.
We revised as follows: We found that EGCG is important for suppression of cognitive decline.
- In the introduction, the following two sentences lack of reference.
“A shorter ingestion period of 2 and 3 months tended to suppress the decrease in learning ability.”
Reference was added.
“The ingestion of EGCG may cause some changes in the brain”.
Without reference, the reader may interpret that those sentences are related to experimental results described in the current paper and not data from the state of the art reported in literature.
This sentence has been changed as follows: It seems that catechin intake needs to be continued for a certain length of time.
- In the introduction, the authors should specify what they mean for “changes in the brain” in the sentence “The ingestion of EGCG may cause some changes in the brain”.
This sentence was ambiguous and has been removed.
- In the results section, in my opinion, the authors should first report the effect of GT-catechins on lifespan and then data on memory acquisition, memory retention and working memory. This is especially important considering that they stated that SAMP10 mice have a median survival time of 10.8 months and they analyzed cognitive performance in 11-mont-old mice.
Thank you very much for your suggestion. We revised the results according to your opinion.
- In the results section (paragraph 2.3), the authors reported transcriptomic analysis to find differentially expressed gene in the hippocampus of SAMP10 mice feed with GT-catechins compared to controls. I was unable to find the indication of mice age as clearly reported for the qPCR analysis. This data is important to compare results from microarray and qPCR approach.
Thank you for pointing out. Added a new sentence as follows: The hippocampus of 2-month-old mice that ingested GT-catechin for one month was used for analysis.
- In the results section (paragraph 2.4), the authors reported that “The increase in IEGs was confirmed by qRT-PCR”. At the same time, they did not confirm the gene expression variation for EGR2 (the transcript with the highest expression variation).
The expression of Egr2 was confirmed by qRT-PCR as shown in figure 4. To our regret, Egr2 was not significantly (p=0.08) but tended to be higher than that of age-matched control.
- In the figure 4, the panel related to Cyr61 contains the indication of control that needs to be removed.
It was removed.
- The figure 5 may be removed considering that no variation has been demonstrated. That data might be reported as “data not shown”.
The figure 5 was removed.
Reviewer 2 Report
The manuscript by Unno et al describes the effects of Green Tea catechins (GT-catechins) on lifespan and cognitive function in the Senescence-accelerated mouse prone 10 (SAMP10) mouse model. Ingestion of GT-catechin improved the cognitive decline and lifespan shortening in the mouse model, in part, through increased expression of IEGs. The study is brief but interesting and the authors’ conclusions are supported by the data shown. The methods are described in sufficient details.
Minor comments:
In the discussion, the authors should include their opinion about the increase in life span after ingestion of GT-catechin in mice. More specifically, how would ingestion of GT-catechin affect lifespan. The authors start out by saying that they want to study pathways other than the suppression of ROS and hence looked at other mechanisms which led to the discovery of increased expression of IEGs. They touch on the topic in the introduction but it would be helpful to know their thoughts about how the life span increase ties in together with their findings.
Author Response
Response to Reviewer 2
The manuscript by Unno et al describes the effects of Green Tea catechins (GT-catechins) on lifespan and cognitive function in the Senescence-accelerated mouse prone 10 (SAMP10) mouse model. Ingestion of GT-catechin improved the cognitive decline and lifespan shortening in the mouse model, in part, through increased expression of IEGs. The study is brief but interesting and the authors’ conclusions are supported by the data shown. The methods are described in sufficient details.
Minor comments:
In the discussion, the authors should include their opinion about the increase in life span after ingestion of GT-catechin in mice. More specifically, how would ingestion of GT-catechin affect lifespan. The authors start out by saying that they want to study pathways other than the suppression of ROS and hence looked at other mechanisms which led to the discovery of increased expression of IEGs. They touch on the topic in the introduction but it would be helpful to know their thoughts about how the life span increase ties in together with their findings.
Thank you for reviewing our manuscript. We added our opinion about the increase in life span to the discussion.
In addition, please find the revised introduction where we added a previous report on the effect of EGCG on nematode longevity.
Reviewer 3 Report
This study examines the effects of GT-catechin on cognitive function and longevitygreen at a wide range of concentrations.
Authors in the introduction section provide the reader with an overview of the scientific topic and the reasons for conducting this research. Author seem to have a good knowledge of the specific research area. In this section, a critical appraisal of previous studies is provided by the authors. The methods section provides a clear description of the experimental procedure. In the Results and Discussion sections the large amount of scientific data are presented in a clear and concise way. These sections highlight significant and interesting findings.
Based on all the above the manuscript can be accepted for publication in Molecules journal.
Author Response
Response to reviewer 3
This study examines the effects of GT-catechin on cognitive function and longevity at a wide range of concentrations.
Authors in the introduction section provide the reader with an overview of the scientific topic and the reasons for conducting this research. Author seem to have a good knowledge of the specific research area. In this section, a critical appraisal of previous studies is provided by the authors. The methods section provides a clear description of the experimental procedure. In the Results and Discussion sections the large amount of scientific data is presented in a clear and concise way. These sections highlight significant and interesting findings.
Based on all the above the manuscript can be accepted for publication in Molecules journal.
Thank you very much for reviewing our manuscript. We appreciate your decision.
Round 2
Reviewer 1 Report
I have a couple of suggestions following reported:
Line 148. I would substitute hippocampi with hippocampus
Line 157. "specifificity", likely a typing error.